# Evaluation of the discriminatory power of spoligotyping and 19-locus mycobacterial interspersed repetitive unit-variable number of tandem repeat analysis (MIRU-VNTR) of *Mycobacterium bovis* strains isolated from cattle in Algeria

Faïza Belakehal[1]*, Stefanie A. Barth[2,3]*, Christian Menge[2], Hamdi T. Mossadak[1], Naïm Malek[4], Irmgard Moser[2,3]

1 High National Veterinary School, Laboratory of Food Hygiene and Quality Insurance System, El-Alia, Oued Smar, Algeria, 2 Friedrich-Loeffler-Institut/Federal Research Institute for Animal Health, Institute of Molecular Pathogenesis, Germany, 3 National Reference Laboratory for Bovine Tuberculosis, at Friedrich-Loeffler-Institut/Federal Research Institute for Animal Health, Institute of Molecular Pathogenesis, Germany, 4 Central Military Hospital, Department of Microbiology, Kouba, Algeria

* faiza.belakehal@gmail.com (FB); stefanie.barth@fli.de (SAB)

## Abstract

Bovine tuberculosis (bTB) caused by *Mycobacterium* (*M.*) *bovis* and *M. caprae* is a transmissible disease of livestock, notifiable to the World Organization for Animal Health (OIE). BTB particularly affects cattle and small ruminants and can be transmitted to humans thereby posing a significant threat to veterinary and public health worldwide. *M. bovis* is the principal cause of bTB in Algeria. In order to better understand the route of spreading and elaborate an eradication program, isolation and characterization of mycobacteria from Algerian cattle was performed. Sixty strains belonging to the *M. tuberculosis* complex were analyzed by spoligotyping, thereof 42 by 19-locus-MIRU-VNTR-typing. Spoligotyping revealed 16 distinguishable patterns (Hunter-Gaston discriminatory index [HGDI] of 0.8294), with types SB0120 (n = 20) and SB0121 (n = 13) being the most frequent patterns, representing 55% of the strains. Analyses based on 19-locus-MIRU-VNTR yielded 32 different profiles, five clusters and one orphan pattern, showing higher discriminatory power (HGDI = 0.9779) than spoligotyping. Seven VNTR-loci [VNTR 577 (alias ETR C), 2163b (QU11b), 2165 (ETR A), 2461 (ETR B), 3007 (MIRU 27), 2163a (QUB11a) and 3232 (QUB 3232)] were the most discriminative loci (HGDI > 0.50). In conclusion, 19-locus-MIRU-VNTR yielded more information than spoligotyping concerning molecular differentiation of strains and better supports the elucidation of transmission routes of *M. bovis* between Algerian cattle herds.

**Data Availability Statement:** All relevant data are within the manuscript and the supporting information.

**Funding:** Parts of this work were funded by a travel and stay grant from the High National Veterinary School, Algeria, (ENSV, www.ensv.dz) (grant no. 000592) for Faiza Belakehal. The funders had no role in study design, data collection and analysis, decision to publish, or preparation of the manuscript.

**Competing interests:** The authors have declared that no competing interests exist.

# Introduction

Bovine tuberculosis (bTB) represents a veterinary and public health threat of global relevance. Despite the importance of the disease, there is no official data concerning the true prevalence of bTB in many low and middle-income countries. This also holds for Algeria, despite the fact that bTB is a notifiable animal disease [1]. Although multiple disease control plans are implemented by the Algerian official's health services, there are still many obstacles to control and eradicate bTB. Skin test and meat inspection at slaughter represent the main methods in bTB surveillance. Application of culture and molecular diagnostic methods is extremely limited in low-income countries due to high costs and time requirements [2]. Several strategies assessing cost effectiveness and socio-economic factors have been proposed to fight bTB [3]. In Algeria, we suspect that most bTB-infected herds are geographically clustered in the north-east of the country because of the high concentration of cattle compared to other areas. It is assumed that transmission of bTB in Algeria is mainly attributed to cattle movement, although a study conducted in France by Palisson [4] disclosed the key role played by the spatial neighborhood of infected herds. Genotyping tools are crucial for unveiling pathogen transmission routes in bTB and tracing back sources of infections. The DR region (direct repeat) analyzed by spoligotyping is a unique region (locus) which contains well-conserved 36 bp direct repeats separated by variable sequences from 35 to 41 bp. These sequences vary from one strain to another by their length, their sequence and their number. The differentiation of the strains is based on the variability of the number of DRs and on the presence or absence of the particular spacers [5]. Because of its simplicity, its binary result format and its high reproducibility, spoligotyping is widely used for the molecular epidemiology of members of the *M. tuberculosis* complex [6]. However, this method is less resolving than either VNTR typing or restriction enzyme analysis, limiting its usefulness in forensic investigations [7]. The VNTR loci containing a variable number of tandem repeats are amplified by PCR using specific primers and the size of the product is determined by gel electrophoresis. The MIRU-VNTR technique is faster than restriction fragment length polymorphism (RFLP) analysis, easy to interpret, more reproducible between laboratories [8] and gives a higher degree of discrimination than spoligotyping [9]. Spoligotyping [5, 10] and mycobacterial interspersed repetitive units-variable number of tandem repeats (MIRU-VNTR) analyses were used [11], since the combination of these two genetic markers is known to be a powerful tool to study the molecular epidemiology in a *M. bovis* population [12]. The main goal of the present study was molecular typing of the Algerian *M. bovis* population in order to improve our understanding of disease epidemiology and bTB spread in Algeria.

# Materials and methods

## Bacterial strains and molecular identification

A total of 3,848 cattle were inspected for bTB at four slaughterhouses in the north part of Algeria (Dellys, El Harrach, Hadjout and Hussein Dey) between January and May 2017. The origin of the animals could not be determined due to the lack of an efficient livestock identification system in Algeria. The most important target organs for tuberculosis lesions, i.e. the respiratory tract (lung tissue and lymph nodes), the thoracic cavity, retropharyngeal lymph nodes, liver and kidney, were inspected at slaughter. Tissue samples were taken during inspection of carcasses for the presence of visible lesions suspected of tuberculosis. The cut surfaces of lymph node were examined carefully for the presence of abscess, caseous mass and tubercles. Out of 3,848 animals screened, 184 carcasses presented with lesions suspicious of bTB and samples from single organs of 105 animals were submitted for microbiological examination.

Organs predominantly affected were respiratory tract and lymph nodes. The tissue samples were decontaminated according to the modified Petroff's method [13, 14]. Briefly, 5 ml of NaOH solution (4%) were added to about 10 g of tissue, minced by a mortar in 4 ml of sterile distilled water, and incubated for 15 min. The suspension was diluted by 10 ml of sterile distilled water and then centrifuged at 3000 x g for 20 min. The pellet was used as an inoculum to solid Löwenstein-Jensen medium. Mycobacteria were cultured at Pasteur Institute of Algiers (IPA), Reference Laboratory for Tuberculosis and Mycobacteria for six to eight weeks according to standard protocols.

The molecular characterization of isolates was performed at the National Reference Laboratory for Bovine Tuberculosis at the Friedrich-Loeffler-Institut, Jena, Germany. Using bacterial DNA extracted by the thermal lysis method described by Berg et al. [15], strains were identified as members of the *M. tuberculosis* complex using the RD9 PCR. The PCR was performed using oligonucleotide primers for detection of the RD9 deletion. A PCR reaction mix consisted of 0.5 μl of each primer (100 μM), 0.2 μl of Taq polymerase (Taq PCR Core Kit, Qiagen, Germany), 2 μl 10x PCR buffer, 4 μl of Q solution, 0.4 μl of 0.25 mM MgCl$_2$, 0.4 μl dNTPs (10 mM each) and 1 μl of purified DNA in a final volume of 20 μl. PCR amplification program included one denaturation step (15 min, 96°C) followed by 35 cycles of 1 min at 95°C, 1 min at 60°C and 1 min at 72°C. PCR amplification products were separated in 1.5% agarose gel after electrophoresis at 110 V for 1 h and visualized using ethidium bromide under UV light. *M. bovis* BCG and *M. tuberculosis* H37Rv strains served as positive controls.

## Molecular typing

Spoligotyping was performed using the DNA microarray format of the ArrayStrip platform (Alere Technologies GmbH [now Abbott], Jena, Germany) according to the manufacturer's instructions as described by Ruettger et al. [10]. Briefly, after DNA extraction with the thermal lysis method, DR regions were amplified using 5-biotinylated primers DRa/DRb [5], and then allowed to hybridize on the microarray (hybridization at 60°C, wash steps at 55°C). Finally, arrays were scanned and analyzed by the ArrayMate™. The Alere Technologies software measures the signal intensity of each probe and compares the profiles automatically to those available at the *M. bovis* Spoligotype Database (www.mbovis.org [16]) and the SITVIT1 database (www.pasteur-guadeloupe.fr:8081/SITVIT_ONLINE/index.jsp [17]).

MIRU-VNTR typing of the *M. bovis* strains was carried out using in-house technique [18], according to Supply et al. [11] with slight modifications, as for VNTR locus 2461-PCR the primers described by Frothingham et al. [19] and for VNTR locus 2163a-PCR the primers of Skuce et al. [20] were used. DNA was purified using the High Pure PCR Template Preparation Kit, Roche Life Science (USA). The PCRs were performed using 0.8 μl of each primer (20 μM; MWG Eurofins, Germany), 0.1 μl of Taq polymerase (HotStarTaq DNA Polymerase, Qiagen, Germany), 2 μl 10x PCR buffer, 4 μl 5x Q solution, 0.4 μl 0.25 mM MgCl$_2$, 0.8 μl dNTPs (10 mM each) and 1 μl of purified DNA in a final volume of 20 μl. PCR amplification program included one denaturation step (15 min, 95°C) followed by 35 cycles of 1 min at 94°C, 1 min at 59°C and 120 s at 72°C. The PCR for VNTR locus 2163a differed with a 25 μl reaction mix (0.5 μl of each primer, 0.1 μl of Hotstar Taq polymerase, 2.5 μl 10x PCR buffer, 1.25 μl 5x Q solution, 0.5 μl dNTPs and 2 μl of purified DNA) and an annealing temperature of 55°C during amplification [20]. PCR amplification products were separated in 1.5% agarose gels after electrophoresis at 100 V for 2 h 15 min and visualized using ethidium bromide under UV light.

Allelic diversity estimations were calculated using the Hunter-Gaston discriminatory index (HGDI) [21]. The HGDI was used to determine the allelic diversity within each MIRU-VNTR

locus and the genotypic diversities (discriminatory power) of the spoligotyping assays, the 19-locus MIRU-VNTR and the combination of both methodologies, using the website application http://insilico.ehu.es/. The sample profiles were analyzed by BioNumerics (version 7.6.2, Applied Maths, The Netherlands) to assess the overall genetic similarity of the strains.

## Results and discussion

### Bacterial strains and molecular identification

Out of 105 samples submitted to microbiological examination, 60 samples showed bacterial growth on Löwenstein-Jensen medium. With a first screening by analysis for acid-fast bacteria, all 60 samples yielded a positive signal and were subsequently subjected to DNA extraction. Subsequent RD9 PCR showed that all 60 strains belonged to *M. tuberculosis* complex whereas non-tuberculous mycobacteria were not isolated.

### Spoligotyping

Of the 60 *M. tuberculosis* complex strains, 59 were identified as *M. bovis* by spoligotyping and one as *M. caprae* (S1 Table). Among the 60 strains, 16 different spoligotypes were identified (Table 1). Four new spoligotypes have been found in the present study and were subsequently submitted to the *M. bovis* spoligotype database (www.mbovis.org). Those new spoligotypes received the ID numbers SB2520 (n = 1), SB2521 (n = 5), SB2522 (n = 1), and SB2523 (n = 1). Two of the four new spoligotypes, SB2520 and SB2521, have been described recently in Algerian cattle and human samples respectively [22].

The predominant spoligotypes found were SB0120 (n = 20) and SB0121 (n = 13), both belonging to the family BOV_1 and differing in only one spacer (no. 21), followed by SB0134 (n = 7), SB2521 (n = 5), and SB0828 (n = 2). The most prevalent spoligotypes SB0120, SB0121,

**Table 1. Spoligotype patterns of 59 *M. bovis* and one *M. caprae* strains and their relative frequencies.**

| SB No. | Spoligotype | | No. of strains | Frequency [%] | SIT | Family |
|---|---|---|---|---|---|---|
| | binary code | octal code | | | | |
| SB0120 | 1101111101111110111111111111111111111100000 | 676773777777600 | 20 | 33.3 | 482 | BOV_1 |
| SB0121 | 1101111101111110111101111111111111111100000 | 676773677777600 | 13 | 21.7 | 481 | BOV_1 |
| SB0134 | 1100011101111110111111111111111111111100000 | 616773777777600 | 7 | 11.7 | 665 | BOV_1 |
| SB2521 | 1100011101111110111011111111111111111100000 | 616773577777600 | 5 | 8.3 | n.k. | n.k. |
| SB0339 | 1101111101111110111101111000011111111100000 | 676773674177600 | 2 | 3.3 | 696 | BOV_1 |
| SB0828 | 1101111101111110111111111111111110111100000 | 676773777773600 | 2 | 3.3 | 1047 | BOV_1 |
| SB1542 | 1100011100011101111111111111111111111100000 | 616173777777600 | 2 | 3.3 | n.k. | BOV |
| SB0119 | 1101111101111001110111111111111111111100000 | 676763677777600 | 1 | 1.7 | 695 | BOV |
| SB0818 | 1101111101111101111111111111111101111100000 | 676773777767600 | 1 | 1.7 | 1044 | BOV_1 |
| SB0822 | 1100111101111110111111111111111111111100000 | 636773777777600 | 1 | 1.7 | 997 | BOV_1 |
| SB0848 | 1101111101110110111011111111111111111100000 | 676733677777600 | 1 | 1.7 | 1010 | BOV_1 |
| SB1452 | 1101111000111001110111111111111111111100000 | 674363677777600 | 1 | 1.7 | 1595 | BOV |
| SB2520 | 1100011101111110111111111110111111111100000 | 616773777377600 | 1 | 1.7 | n.k. | n.k. |
| SB2522 | 1101111101111101111100111111111011111100000 | 676773717757600 | 1 | 1.7 | n.k. | n.k. |
| SB2523 | 1101111101111101001111111111111111111100000 | 676772377777600 | 1 | 1.7 | n.k. | n.k. |
| SB0835 | 0100000000000001111111111101110011111100000 | 200003777357600 | 1 | 1.7 | 978 | BOV_4 Caprae |
| | Total | | 60 | 100 | | |

n.k.: not known; **SIT**: spoligo-international type according to the SITVIT1 nomenclature; **BOV**: Bovine.

The binary code is represented by 43 digits. The number 0 indicate the absence of signal and the number 1 indicate the presence of signal.

and SB0134 were already detected by a study conducted 10 years earlier in Algeria [23]. Of note, the two new types SB2522 and SB2523 as well as three additional spoligotypes, namely SB0119 (n = 1), SB0848 (n = 1), and SB1542 (n = 2) have not been reported in Algeria before.

Presence of spoligotypes SB0121, SB0120, and SB0134 was also reported for the neighboring countries Morocco and Tunisia as well as France [24–26]. Interestingly, the second most frequently found spoligotype in Morocco, type SB0265, was not detected in our study in Algerian cattle although in silico analysis of whole genome sequence (WGS) data from four Algerian human *M. bovis* isolates [27] could assign two of the strains to SB0265 implying that the spoligotype is present in Algeria, too. The BCG cluster associated spoligotypes SB0120 and SB0121 are circulating nearly worldwide in Europe, Africa, and America [28, 29]. Spoligotype SB0134 is also frequent in many African countries (e.g., Tunisia, Morocco, Mali and Ethiopia) [30–33]. The new type SB2520 identified in our study has been recently found in Ethiopia [33]. The spoligotypes SB0339 and SB0119 are common on the Iberian peninsula, thereof SB0339 frequently reported for Spain and SB0119 in Portugal, and both were also reported for Morocco [24, 34, 35]. The spoligotype SB0818 has been reported in Italy [36] and France [26, 37], SB0848 in the neighboring country Tunisia [31] and in Portugal [38]. According to the database www.mbovis.org, spoligotype SB1542 was previously detected in Italy and *M. caprae* SB0835 in France. *M. caprae* has been identified mainly in continental Europe in the alpine region [39] and Spain [40]. In the study presented here only one *M. caprae* isolate with the spoligotype SB0835 was detected. Only few data exist on the detection of *M. caprae* in Africa. In Algeria two isolates of spoligotypes SB1451 and SB0835 were described before [22, 23], pointing to a rare occurrence of *M. caprae* in this country.

Nowadays, the composition of the spoligotypes in North Africa (Morocco, Algeria, Tunisia) is dominated by spoligotypes SB120, SB121 and SB0134 possibly having evolved from common ancestral strains introduced since decades from Europe [23–25]. Indeed, this can be explained by recent imports, given the incomplete eradication of bTB in Europe. France represents the country from which the majority of cattle in Algeria were imported. According to WAHIS interface, France is declared "officially tuberculosis free" (OTF) by the EU but bTB still exists and single outbreaks are repeatedly reported for cattle herds and wildlife in some areas of the country. Thereby, SB0120, SB121 and SB0134 are the major spoligotypes distributed in animals in France, too [26]. Moreover, a study conducted in Tunisia by Siala and colleagues analysed human *M. bovis* isolates and found 13 diffferent spoligotypes, thereof, SB0120 and SB0121 as the dominating types [41].

## MIRU-VNTR typing

Out of 60 strains of this study, VNTR profiles could be generated for 42 *M. bovis* strains. Missing success in the remaining cases might have been due to low DNA quality or quantity. A panel of 19 loci was chosen for conducting the MIRU-VNTR (Table 2).

Evaluated loci were selected according to the 12-locus panel proposed in the MIRU-VNTR plus web site (www.miru-vntrplus.org) in combination with the panel proposed by the European Reference Center for Bovine Tuberculosis (EU-RL bTB, VISAVET Health Surveillance Centre, Universidad Complutense de Madrid) for the molecular typing of *M. bovis* in Europe. Among them, the ETR loci (VNTR580, 2165, 2461, and 3192) had been evaluated by Sahraoui et al. [42] for Algerian strains previously. Additionally, the QUB loci (VNTR2163a, 2163b, 3232, and 4052) and VNTR2996 were included as these are regarded highly discriminative for *M. bovis* strains.

Overall, seven loci (VNTR577, 2163a, 2163b, 2165, 2461, 3007, and 3232) presented a high discriminatory power (HGDI > 0.50, Table 2). Our findings corroborate earlier data obtained

**Table 2. Allele diversity of the 19-locus MIRU-VNTR in 42 Algerian *M. bovis* strains.**

| Locus | | Number of strains with the respective number of copies | | | | | | | | | | | | | Allele diversity (HGDI) |
|---|---|---|---|---|---|---|---|---|---|---|---|---|---|---|---|
| VNTR | Alias | 1 | 2 | 3 | 4 | 5 | 6 | 7 | 8 | 9 | 10 | 11 | 12–15 | 16 | |
| 2165 | ETR-A | 0 | 0 | 1 | 7 | 11 | 15 | 8 | 0 | 0 | 0 | 0 | 0 | 0 | 0.7573 |
| 3232 | QUB3232 | 0 | 0 | 2 | 1 | 0 | 17 | 13 | 8 | 1 | 0 | 0 | 0 | 0 | 0.7178 |
| 2163a | QUB11a | 2 | 0 | 0 | 0 | 1 | 0 | 0 | 0 | 15 | 17 | 5 | 0 | 1 | 0.6927 |
| 2163b | QUB11b | 0 | 16 | 8 | 16 | 1 | 0 | 1 | 0 | 0 | 0 | 0 | 0 | 0 | 0.6887 |
| 2461 | ETR-B | 0 | 0 | 1 | 20 | 19 | 2 | 0 | 0 | 0 | 0 | 0 | 0 | 0 | 0.5796 |
| 3007 | MIRU27 | 3 | 17 | 22 | 0 | 0 | 0 | 0 | 0 | 0 | 0 | 0 | 0 | 0 | 0.5703 |
| 577 | ETR-C | 1 | 1 | 13 | 2 | 25 | 0 | 0 | 0 | 0 | 0 | 0 | 0 | 0 | 0.5598 |
| 4052 | QUB26 | 3 | 1 | 2 | 5 | 30 | 1 | 0 | 0 | 0 | 0 | 0 | 0 | 0 | 0.4785 |
| 3192 | MIRU31/ETR-E | 0 | 3 | 33 | 6 | 0 | 0 | 0 | 0 | 0 | 0 | 0 | 0 | 0 | 0.3659 |
| 580 | MIRU04/ETR-D | 2 | 0 | 38 | 1 | 0 | 0 | 0 | 0 | 1 | 0 | 0 | 0 | 0 | 0.1823 |
| 2687 | MIRU24 | 2 | 40 | 0 | 0 | 0 | 0 | 0 | 0 | 0 | 0 | 0 | 0 | 0 | 0.0929 |
| 154 | MIRU02 | 1 | 41 | 0 | 0 | 0 | 0 | 0 | 0 | 0 | 0 | 0 | 0 | 0 | 0.0476 |
| 802 | MIRU40 | 1 | 41 | 0 | 0 | 0 | 0 | 0 | 0 | 0 | 0 | 0 | 0 | 0 | 0.0476 |
| 2059 | MIRU20 | 1 | 41 | 0 | 0 | 0 | 0 | 0 | 0 | 0 | 0 | 0 | 0 | 0 | 0.0476 |
| 2996 | MIRU26 | 0 | 0 | 0 | 1 | 41 | 0 | 0 | 0 | 0 | 0 | 0 | 0 | 0 | 0.0476 |
| 4348 | MIRU39 | 0 | 41 | 1 | 0 | 0 | 0 | 0 | 0 | 0 | 0 | 0 | 0 | 0 | 0.0476 |
| 960 | MIRU10 | 0 | 42 | 0 | 0 | 0 | 0 | 0 | 0 | 0 | 0 | 0 | 0 | 0 | 0 |
| 1644 | MIRU16 | 0 | 0 | 42 | 0 | 0 | 0 | 0 | 0 | 0 | 0 | 0 | 0 | 0 | 0 |
| 2531 | MIRU23 | 0 | 0 | 0 | 42 | 0 | 0 | 0 | 0 | 0 | 0 | 0 | 0 | 0 | 0 |

**ETR:** Exact Tandem Repeat; **MIRU:** Mycobacterial Interspersed Repetitive Unit; **QUB:** Queen's University Belfast.

in Tunisia [25] which showed a high resolution of VNTR2163a, 2163b, 2461, 2165, and 3232, while the four loci VNTR577, 2165, 2461, and 3007, showed a high discriminatory power for Algerian strains also in previous data sets [42]. Furthermore, this is consistent with data reported for *M. bovis* strains from Mexico and Germany [43, 44]. Rodriguez-Campos et al. [45] found only a very low allele diversity for locus VNTR2163b (HGDI = 0.08), but analyzed only SB0121 strains. The two loci VNTR3192 and 4052 showed moderate allelic diversity (HGDI = 0.36 and HGDI = 0.47, respectively) and locus VNTR580 a low allelic diversity (HGDI = 0.18). Very low allele diversity (HGDI <0.15) was observed for six loci (VNTR154, 802, 2059, 2687, 2996, and 4348). By contrast, Yang et al. [46], found a high allele diversity in the loci VNTR580 (HGDI = 0.607) and 802 (HGDI = 0.495) but no resolution for loci VNTR2059, 2687 and 2996. This difference may be related especially to the geographic region from which the strains have been isolated, as we analyzed exclusively bovine strains from Algeria whereas Yang et al. typed strains from Sika deer in Northern China [46]. Finally, there is a difference in the resolution of loci between *M. tuberculosis* complex species. For example, loci VNTR960 and 2531 are reported very discriminatory for *M. tuberculosis* [47], but performed less well for *M. bovis* in the present study. VNTR2996 demonstrates a very low allele diversity in the present study (HGDI = 0.047). A possible explanation could be, that the number of strains analyzed in the present study was smaller and therefore less variance found. No allele diversity (HGDI = 0) was observed with the loci VNTR960, 1644, and 2531 which suggests that these loci are not suitable for typing Algerian *M. bovis* strains. Overall, analysis of 19-locus MIRU-VNTR in 42 strains revealed 32 different profiles among which 27 were found only in a single strain (**Table 3**).

The combination of MIRU-VNTR allele diversity and spoligotyping pattern demonstrated a large genotypic variety resulting in 33 profiles (Fig 1). Thereby spoligotypes were typed in

**Table 3. Comparison of the discriminatory power of spoligotyping, 19-locus MIRU-VNTR typing and the combination of both methods.**

| Variability | Spoligotyping | MIRU-VNTR | Spoligotyping and MIRU-VNTR |
|---|---|---|---|
| No. of strains included | 60 | 42 | 42 |
| Total number of profiles (n) | 16 | 32 | 33 |
| Number of individual profiles (n) | 9 | 27 | 28 |
| Discriminatory index (HGDI) | 0.8294 | 0.9779 | 0.9826 |

sub-profiles, e.g. SB0120 in 13 sub-profiles, SB0121 in five, and SB0134 in five, reflecting the heterogeneity of strains causing bTB in Algeria.

Scientific literature describes several routes of transmission of animal tuberculosis [48, 49]. The emergence of tuberculosis in wildlife [50] may constitute a continuous source for reinfection of cattle and could be at the origin of the persistence of bTB worldwide. This particularly

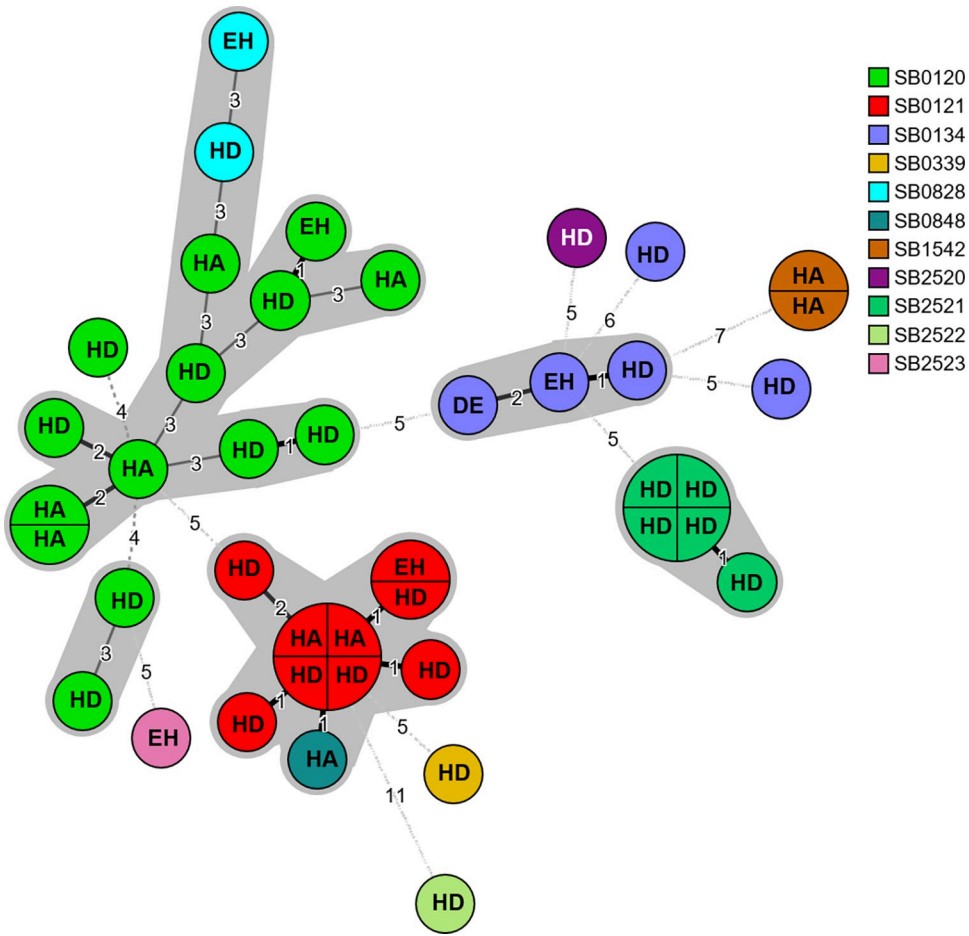

**Fig 1. Minimum-spanning-tree based on the combination of spoligotyping and MIRU-VNTR results of 42 *M. bovis* strains isolated from slaughtered cattle in Algeria.** Each node represents a unique spoligotype-VNTR sub-profile. If more than one strain exhibited the identical profile the node is separated. Nodes closer than three differences are put into one cluster (gray background). The number of differing alleles (repeat number of MIRU-VNTR and absence/presence of spacer for spoligotyping) between two nodes is indicated by the respective number. The letter inside the nodes represent the slaughterhouse the strains came from (**HD**: Hussein Dey, **HA**: Hadjout, **EH**: El Harrach, **DE**: Dellys) (BioNumerics, version 7.6.2).

holds for wild boar, which represent a new threat for livestock animals and agriculture in Algeria. Hence, the pathway of transmission of this pathogen in Algeria including wildlife needs to be studied. El Mrini et al. [51] described cases of tuberculosis due to *M. bovis* in Eurasian wild boar in Morocco. Without neglecting the role of re-infections within single cattle herds, mycobacteria may survive decades unrecognized in several biological niches due to the chronic sub-clinical course of infection, leading to persisting bTB in Algerian livestock. On the other hand, all our *M. bovis* strains were recovered from lungs and associated lymph nodes, suggesting a primarily airborne transmission of bTB to the cattle under study here and, consequently, an intimate contact between the respective animals. Indeed, combination of MIRU-VNTR allele diversity and spoligotyping pattern allowed us to unveil some possible epidemiological links. Constructing a minimum-spanning-tree revealed five clusters (spoligotype and MIRU-VNTR profile differing in 3 allels in maximum) with the largest one comprising of 14 strains. Two sub-profiles (one in the SB0121 cluster and the other in the SB2521 cluster) contained four strains, that were identical in their spoligotype and MIRU-VNTR profiles, two additional sub-profiles were represented by two strains each. The four strains of the SB2521 sub-profile were all isolated at the slaughterhouse Hussein Dey. It is tempting to assume that cross-infection occurred between animals within one herd or the same area. In contrast, both sub-profiles with four and two strains, respectively, within the spoligotype SB0121 contained strains probably from different areas in Algeria as the strains were isolated from cattle slaughtered in different abattoirs. This includes the slaughterhouses of Hussein Dey and Hadjout which are 88 km distant from each other (source of the strains belonging to the four strain sub-profile), as well as slaughterhouses in El Harrach and Hussein Dey which are more proximate to each other. As we consider it unlikely that one owner had delivered his cattle to different slaughterhouses, the occurrence of identical spoligotype/MIRU-VNTR profiles in different geographical areas might be explained by the trade of animals between herds in those areas. In two cluster, two different spoligotypes (SB0848 and SB0121 as well as SB0828 and SB0120) are combined. Both pairs differ from each other only in one spacer (no. 13 is missing in SB0848 compared to SB0121 as well as no. 34 is missing in SB0828 compared to SB0120) it might be possible that spoligotypes SB0828 and SB0848 may have originated from spoligotype SB0120 and SB0121, respectively, by undergoing a genetic mutation on spacer 13 respectively 34. The singular SB0848 strain and four SB0121 strains possess identical MIRU-VNTR profiles, supporting the close relatedness of both spoligotypes.

Beyond disclosing recent transmission events and microevolution of *M. bovis* strains, genotyping of bacteria also may allow for a reconstruction of phylogeography and evolutionary history of these pathogens [52]. Spoligotyping in combination with other genomic markers, such as deletions [52, 53] and single-nucleotide polymorphisms (SNPs) have been used to construct a phylogeography for *M. bovis* strains. The clonal complexes named "African 1" (Af1, characterized by missing of spacer 30) and "African 2" (Af2, missing spacers 3–7) were described as geographically localized in Central-West and East Africa, respectively. The cluster "European 1" (Eu1, missing spacer 11) is found mainly on the British Islands, in New Zealand, Australia, South Africa, and Korea and "European 2" (Eu2, missing spacer 21) on the Iberian peninsula and Western Europe [52–54]. Additional to those four 'classical' phylogenetic lineages recently eight new clonal lineages were defined on the basis of WGS data of more than 3,300 *M. bovis* isolates and spoligotypes where affiliated to certain phylogenetic lines [52]. Six out of the 16 spoligotypes detected in our study are also mentioned in this recent publication. The discriminatory capacity of spoligotyping is limited since diversity is measured at a single locus prone to convergent evolution and phylogenetic distances cannot be reliably inferred [55]. Despite this limitation our findings might point towards the phylogenetic positions of strains and indicate the multiplicity of clonal lines circulating in Algeria. The signature profile of the *M. bovis* BCG

vaccine strain is the absence of spacers 3, 9, 16, and 39 to 43 [29]. The BCG cluster is divided into two groups: the BCG-like group, represented by SB0121, and the ancestor BCG-like SB0120. In the present study, all *M. bovis* strains lacked spacers 3, 9, 16, and 39 to 43 and were therefore grouped to the BCG cluster (Table 1). On the African continent, the ancestral BCG-like cluster is predominantly found in Algeria, Zambia [56], and Mozambique [57]. The cluster Af1, Af2, and Eu1 have not been found in cattle outside of the previous mentioned regions [53]. Further studies, implicating WGS technology, will be required to determine if or to what extend the current bTB epizootic in Algeria originates from the long-term evolution during which animal-adapted *M. tuberculosis* complex, i.e. *M. caprae* and *M. bovis*, might have originally come out of Africa [58] or on a more recent (re)introduction of *M. bovis* from Europe as discussed above.

## Conclusion

Until now, tuberculosis still represents a serious burden in Algeria for both, humans and animals. The BCG cluster derived spoligotypes (SB0120, SB0121) are the most frequent types circulating in Algeria and also worldwide. The analysis of four MIRU-VNTR loci [ETR A, VNTR2163b (QUB11b), 2163a (QUB11a), and 3232 (QUB3232)] would be sufficient to characterize Algerian *M. bovis* strains deeply. Nevertheless, the combination of MIRU-VNTR and spoligotyping is highly advisable. The association of those two techniques demonstrates the heterogenic population of Algerian *M. bovis* strains, indicating that different strains are responsible for bTB in Algeria. However, the lack of information concerning the origin of cattle (herd, country) currently impairs the unveiling of transmission routes of bTB in Algeria and must be implemented as part of the control program for the eradication of bovine tuberculosis in Algeria.

## Supporting information

**S1 Table. Spoligotyping pattern and MIRU-VNTR profile of 59 *Mycobacterium bovis* and 1 *Mycobacterium caprae* strain from cattle in Algeria.** Abbreviations: nt, not tested; neg, no PCR fragment detectable.
(XLSX)

## Acknowledgments

The author's address special thanks to the technicians Uta Brommer and Josefine Bach at FLI, Jena, Germany, for their excellent support to perform the molecular methods. We are grateful to Prof. Dr. Dr. h.c. Thomas C. Mettenleiter for providing the opportunity to perform the practical part of this work at the FLI, Jena, Germany.

## Author Contributions

**Conceptualization:** Faïza Belakehal, Hamdi T. Mossadak, Naïm Malek, Irmgard Moser.

**Data curation:** Stefanie A. Barth.

**Formal analysis:** Faïza Belakehal, Stefanie A. Barth.

**Investigation:** Faïza Belakehal.

**Methodology:** Stefanie A. Barth, Irmgard Moser.

**Supervision:** Stefanie A. Barth, Hamdi T. Mossadak.

**Visualization:** Stefanie A. Barth.

**Writing – original draft:** Faïza Belakehal.

**Writing – review & editing:** Faïza Belakehal, Stefanie A. Barth, Christian Menge, Irmgard Moser.

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
