## [Decision Letter · Decision Letter 0]

24 Nov 2021

PONE-D-21-26506Evaluation of the discriminatory power of spoligotyping and 19-locus mycobacterial interspersed repetitive unit-variable number of tandem repeat analysis (MIRU-VNTR) of Mycobacterium bovis strains isolated from cattle in AlgeriaPLOS ONE

Dear Dr. Barth,

Thank you for submitting your manuscript to PLOS ONE. After careful consideration, we feel that it has merit but does not fully meet PLOS ONE’s publication criteria as it currently stands. Therefore, we invite you to submit a revised version of the manuscript that addresses the points raised during the review process. Please see comments made by the reviewer (see below this letter), and also please note that there is a PDF file with reviewer's comments made directly in the file. I ask you to revise your paper accordingly. Please submit your revised manuscript by Jan 08 2022 11:59PM. If you will need more time than this to complete your revisions, please reply to this message or contact the journal office at plosone@plos.org. Please include the following items when submitting your revised manuscript:A rebuttal letter that responds to each point raised by the academic editor and reviewer(s). You should upload this letter as a separate file labeled 'Response to Reviewers'.A marked-up copy of your manuscript that highlights changes made to the original version. You should upload this as a separate file labeled 'Revised Manuscript with Track Changes'.An unmarked version of your revised paper without tracked changes. You should upload this as a separate file labeled 'Manuscript'.If applicable, we recommend that you deposit your laboratory protocols in protocols.io to enhance the reproducibility of your results. Protocols.io assigns your protocol its own identifier (DOI) so that it can be cited independently in the future. For instructions see: https://journals.plos.org/plosone/s/submission-guidelines#loc-laboratory-protocols. Additionally, PLOS ONE offers an option for publishing peer-reviewed Lab Protocol articles, which describe protocols hosted on protocols.io. Read more information on sharing protocols at https://plos.org/protocols?utm_medium=editorial-email&utm_source=authorletters&utm_campaign=protocols.

We look forward to receiving your revised manuscript.

Kind regards,

Igor Mokrousov, Ph.D., D.Sc.

Academic Editor

PLOS ONE

Journal Requirements:

2. In your Methods section, please provide the name of the slaughterhouse where the animals were sacrificed.

3. We note that you are reporting an analysis of a microarray, next-generation sequencing, or deep sequencing data set. PLOS requires that authors comply with field-specific standards for preparation, recording, and deposition of data in repositories appropriate to their field. Please upload these data to a stable, public repository (such as ArrayExpress, Gene Expression Omnibus (GEO), DNA Data Bank of Japan (DDBJ), NCBI GenBank, NCBI Sequence Read Archive, or EMBL Nucleotide Sequence Database (ENA)). In your revised cover letter, please provide the relevant accession numbers that may be used to access these data. For a full list of recommended repositories, see http://journals.plos.org/plosone/s/data-availability#loc-omics or http://journals.plos.org/plosone/s/data-availability#loc-sequencing.

Reviewers' comments:

Reviewer's Responses to Questions

**Comments to the Author**

1. Is the manuscript technically sound, and do the data support the conclusions?

Reviewer #1: Yes

2. Has the statistical analysis been performed appropriately and rigorously? 

Reviewer #1: Yes

3. Have the authors made all data underlying the findings in their manuscript fully available?

Reviewer #1: Yes

4. Is the manuscript presented in an intelligible fashion and written in standard English?

Reviewer #1: Yes

5. Review Comments to the Author

Reviewer #1: The manuscript results presented the most frequent M. bovis strains in Algeria. Characterisation of M. bovis strains is important in tracing transmission of the disease. However, the data was incomplete in terms of tracing the origin of the cattle as this would be important in revealing transmission routes, hence implementation of control methods.

The manuscript was well written and clear. The authors described the laboratory procedures well and the methods of data interpretation were appropriate. There was no need for statistical analysis.

6. PLOS authors have the option to publish the peer review history of their article (what does this mean?). If published, this will include your full peer review and any attached files.

Reviewer #1: **Yes: **Petronillah Manhondo

---

## [Author Response · Author response to Decision Letter 0]

21 Dec 2021

1. Please ensure that your manuscript meets PLOS ONE's style requirements, including those for file naming. The PLOS ONE style templates can be found at https://journals.plos.org/plosone/s/file?id=wjVg/PLOSOne_formatting_sample_main_body.pdf and https://journals.plos.org/plosone/s/file?id=ba62/PLOSOne_formatting_sample_title_authors_affiliations.pdf.

We updated the title page and the style of the headings in the manuscript in order to meet all requirements.

2. In your Methods section, please provide the name of the slaughterhouse where the animals were sacrificed.

The name of the corresponding slaughterhouse has been provided in the method section of the paper as well as the supporting information S1 Table. 

3. We note that you are reporting an analysis of a microarray, next-generation sequencing, or deep sequencing data set. PLOS requires that authors comply with field-specific standards for preparation, recording, and deposition of data in repositories appropriate to their field. Please upload these data to a stable, public repository (such as ArrayExpress, Gene Expression Omnibus (GEO), DNA Data Bank of Japan (DDBJ), NCBI GenBank, NCBI Sequence Read Archive, or EMBL Nucleotide Sequence Database (ENA)). In your revised cover letter, please provide the relevant accession numbers that may be used to access these data. For a full list of recommended repositories, see http://journals.plos.org/plosone/s/data-availability#loc-omics or http://journals.plos.org/plosone/s/data-availability#loc-sequencing.

We included all data generated during this study in the manuscript in supplemental Table S1. 

We carefully revised our reference list and found no retracted manuscript. For some references we updated the pmid no. In page 17, Line 316 we replaced reference number 24 by number 53.

+++++++++++++++++++++++++++++++

Reviewers' comments:

>Page 8, Line 35: delete has been and replace with was.

>Page 10, Line 93: replace with were

>Page 10, Line 96: Replace to with for

>Page 11, Line 98: Delete an amount of and statement should read Briefly, 5ml of NaOH.......

>Page 11, Line 113: Delete s

>Page 11, Line 114-116: Rephrase statement not making sense M. bovis ( ) and M. tuberculois ( ) strains served as positive controls. 

>Page 12, Line 127: Replace by with using

>Page 13, Line 150: Delete and replace with Out of 

>Page 13, Line 156: Remove, and add and 

>Page 15, Line 169: delete

>Page 18, Line 223: Delete therefore

We changed the text in all sites of the manuscript as proposed by the reviewer.

>Page 9, Line 65-67: This statement should be in the last paragraph at the end of the Introduction.

>Page 9, Line 67-69: Added at the end of the introduction as part of the objectives description.

We moved both sentences to the end of the introduction paragraph.

>Page 11, Line 106 : Berg et al. [15]

For reference no. 15 (Berg et al.) we added the doi of the Manuscript Correction in the reference list. 

>Page 11, Line 120: Ruettger et al. [6]

For reference no. 6 (Ruettger et al.) it was not clear to us what the reviewer would have liked to changed, as this paper describes the used spoligotyping method in detail.

---

## [Editor Report · Decision Letter 1]

23 Dec 2021

Evaluation of the discriminatory power of spoligotyping and 19-locus mycobacterial interspersed repetitive unit-variable number of tandem repeat analysis (MIRU-VNTR) of Mycobacterium bovis strains isolated from cattle in Algeria

PONE-D-21-26506R1

Dear Dr. Barth,

We’re pleased to inform you that your manuscript has been judged scientifically suitable for publication and will be formally accepted for publication once it meets all outstanding technical requirements.

Kind regards,

Igor Mokrousov, Ph.D., D.Sc.

Academic Editor

PLOS ONE